# Single crystalline quaternary sulfide nanobelts for efficient solar-to-hydrogen conversion

Liang Wu[1,2,5], Qian Wang[3,5], Tao-Tao Zhuang[1,2], Yi Li[1,2], Guozhen Zhang [3], Guo-Qiang Liu[1,2], Feng-Jia Fan[4], Lei Shi[1] & Shu-Hong Yu [1,2 ✉]

Although solar-driven water splitting on semiconductor photocatalysts is an attractive route for hydrogen generation, there is a lack of excellent photocatalysts with high visible light activity. Due to their tunable bandgaps suitable for superior visible-light absorption, copper-based quaternary sulfides have been the important candidates. Here, we first assessed the preferred facet of wurtzite Cu-Zn-In-S for photocatalytic hydrogen evolution reaction using the relevant Gibbs free energies determined by first principle calculation. We then developed a colloidal method to synthesize single crystalline wurtzite Cu-Zn-In-S nanobelts (NBs) exposing (0001) facet with the lowest reaction Gibbs energy, as well as Cu-Zn-Ga-S NBs exposing (0001) facet. The obtained single crystalline Cu-Zn-In-S and Cu-Zn-Ga-S NBs exhibit superior hydrogen production activities under visible-light irradiation, which is composition-dependent. Our protocol represents an alternative surface engineering approach to realize efficient solar-to-chemical conversion of single crystalline copper-based multinary chalcogenides.

[1] Division of Nanomaterials & Chemistry, Hefei National Laboratory for Physical Sciences at the Microscale, University of Science and Technology of China, 230026 Hefei, China. [2] Institute of Energy, Hefei Comprehensive National Science Center, CAS Center for Excellence in Nanoscience, Department of Chemistry, Institute of Biomimetic Materials & Chemistry, University of Science and Technology of China, 230026 Hefei, China. [3] Department of Chemical Physics, iChEM (Collaborative Innovation Center of Chemistry for Energy Materials), Hefei National Laboratory for Physical Sciences at the Microscale, University of Science and Technology of China, 230026 Hefei, Anhui, China. [4] CAS Key Laboratory of Microscale Magnetic Resonance and Department of Modern Physics, Synergetic Innovation Center of Quantum Information and Quantum Physics, University of Science and Technology of China, 230026 Hefei, Anhui, China. [5] These authors contributed equally: Liang Wu, Qian Wang. ✉ email: shyu@ustc.edu.cn

The constantly increasing global energy crisis and related environmental problems promote researchers to find renewable and ecofriendly energy sources. A promising candidate is clean hydrogen energy[1–3]. Among all methods for producing hydrogen, photocatalytic water splitting using semiconductor nanomaterials is one of most efficient ways[4–9]. However, the wide optical band gap and intrinsic toxicity of the most photocatalysts have impeded the practical, complete, and renewable solar-driven hydrogen production process. Thus, developing an efficient, non-toxic photocatalyst with wide optical absorption region is still an important issue, which is highly needed to be addressed.

Copper-based quaternary sulfide nanomaterials, especially for Cu-Zn-In-S (CZIS) and Cu-Zn-Ga-S (CZGS), which consist of non-toxic and earth abundant elements are attractive candidate for solar-to-hydrogen conversion because of their tunable bandgap, environmental benignity, good thermal and chemical stability, and easy synthesis from abundant and inexpensive precursors[10–16]. Since Domen and co-workers reported CZIS enabled photocatalytic hydrogen production aided by co-catalyst, many efforts have been paid on developing copper-based multinary sulfide photocatalysts, which hold great potentials in solar energy conversion and chemical synthesis[17–21].

However, the low electric conductivity, rapid recombination rate of photogenerated holes and electrons, and the less accessible surface active sites are adverse to their photocatalytic performance. Although loading noble metals and constructing heterostructures are effective ways to enhance the photocatalytic performances of copper-based multinay sulfides[22–24], there are some drawbacks remaining, such as the high cost and poor interfacial interaction in heterojunction. Furthermore, tailoring the morphology and surface facets of semiconductors can efficiently enhance and optimize the photocatalytic hydrogen evolution performance[25–33]. Moreover, well-defined single crystalline CZIS will be free of grain boundaries and defects, which are recombination and trapping centers for photogenerated electrons and holes[34,35]. Therefore, controlling synthesis of single-crystalline wurtzite CZIS nanostructures with a special surface crystal facet which is the best for photocatalytic hydrogen production can efficiently enhance their photocatalytic properties.

Here, we first identify that the (0001) facet of wurtzite CZIS has the smallest Gibbs free energy change for photocatalytic hydrogen evolution reaction (HER) using first principle density functional theory (DFT) calculation. According to the Bell-Evans-Polanyi Principle, the (0001) facet possesses the lowest energy barrier for HER, which would facilitate the hydrogen production. We then design a simple colloidal method to synthesize single crystalline wurtzite CZIS nanobelts (NBs) exposing the (0001) facet, as well as the single crystalline wurtzite CZGS NBs with the exposed (0001) facet assisted with oleylamine (OLA) and 1-dodecanethiol (DDT). The as-prepared nanobelt photocatalysts show excellent composition-dependent photocatalytic performances, reaching the highest hydrogen production rate of 3.35 and 3.75 mmol h$^{-1}$ g$^{-1}$, respectively, for CZIS and CZGS nanobelts under visible-light irradiation ($\lambda > 420$ nm) without co-catalyst, which are higher than the reported CZIS and CZGS photocatalysts (Supplementary Table 1). Importantly, the NBs keep high stability and catalytic reactivity after storing two months, further proving that constructing two-dimensional materials is beneficial for prospective advanced applications. The excellent photocatalytic performance of CZIS and CZGS NBs demonstrated here pave the way for facet engineering of semiconductor photocatalysts in the future.

## Results
### Crystal facet screening using theoretical simulation.
Copper-based multinary chalcogenides have been widely utilized as photocatalysts for solar-to-hydrogen generation[10,36,37]. Among them, Cu-Zn-In-S and Cu-Zn-Ga-S can be investigated as a new kind of catalyst due to their tunable bandgap and suitable absorption region[15,16]. Taken wurtzite Cu-Zn-In-S as an example, the crystal structure can be obtained through replacing Zn with Cu and In atoms in wurtzite ZnS (Fig. 1a)[38,39]. To clarify the influence of the exposed surface facets of wurtzite CZIS nanocrystals on photocatalytic hydrogen evolution, we used DFT calculation to explore the reaction Gibbs energy ($\Delta G_H$) of (0001), (1010), and (1011) facets (Supplementary Fig. 1). As seen from Supplementary Fig.2, the band gap of CZIS predicted by HSE06 (2.0 eV) agrees well with the experimentally measured value (2.1 eV) (Supplementary Table 2). The H absorption energy of different sites in the selected facets (Supplementary Fig. 3 and Supplementary Table 3) reveal that the (0001) facet provide a favorable environment for hydrogen production. The calculated $\Delta G_H$ of different facets (Fig. 1b) show that the (0001) facet has the smallest binding strength (0.589 eV) to atomic hydrogen, comparing to (1010) facets (1.506 eV) and (1011) facets (1.549 eV). Following the Bell–Evans–Polanyi principle, we expect that the (0001) facet is the most favorable surface for photocatalytic hydrogen production on CZIS, which has inspired us to design CZIS nanocrystals with (0001) facet exposed for HER.

### Colloidal synthesis of single-crystalline wurtzite CZIS NBs.
Therefore, we sought to synthesize a wurtzite CZIS photocatalyst just exposing the (0001) facet, that is 2D nanostructure with a (0001) facet surface. At present, major copper-based quternary sulfide nanocrystals are synthesized in spherical and rod-like morphology through colloidal method[40–44]. As schematically illustrated in Supplementary Fig. 4 (The detail procedure is displayed in experiment section), the ultrathin single crystalline CZIS nanobelts were prepared via a facial colloidal method, which is assisted by OLA and DDT ligands. A number of factors (reaction temperature, reaction time, the kinds, and ratio of ligands) have been investigated to optimize the synthesis condition to obtain two-dimensional CZIS (Supplementary Figs. 5–8).

We further study the crystal structure of the synthesized CZIS nanobelts. As shown in Fig. 1c, the X-ray diffraction (XRD) pattern shows a typical hexagonal wurtzite structure diffraction peak similar to the reported wurtzite CZIS and CZTS[45,46], which agrees well with the simulated wurtzite CZIS shown below for reference, and no other phases are detected. Both the calculated structure in terms of lattice constants and the simulated XRD data based on it agree well with the corresponding experimental data (Supplementary Table 4). When the colloidal suspensions of CZIS nanobelts are drop-cast onto planar substrates, they show specific orientation as indicated by the obvious (0002) diffraction peak in the XRD pattern (Fig. 1c). This result indicates that the products consist almost all of [0001]-oriented nanobelts, indirectly proving the 2D structure with the exposed (0001) facet[47]. Besides, the oxidation state and composition of the surface elements of the obtained CZIS nanobelts were characterized by X-ray photoelectron spectroscopy (XPS; Supplementary Fig. 9). The survey spectrum shows the existence of Cu (I), In (III), Zn (II), and S (II) states in the CZIS nanobelts[48,49].

Furthermore, the morphology of the synthesized CZIS NBs was investigated by the transmission electron microscopy (TEM) and high-angle annular dark field scanning transmission electron microscopy (HAADF-STEM). Figure 1d, e display a 2D nanobelt structure with a nearly transparent characteristic, meaning the ultrathin nature of the synthesized CZIS nanobelts. Then, atomic force microscope (AFM) was used to characterize the thickness of the synthesized nanobelts. The AFM image and statistical height profiles (Fig. 1f, g) show that the thinnest CZIS nanobelts present

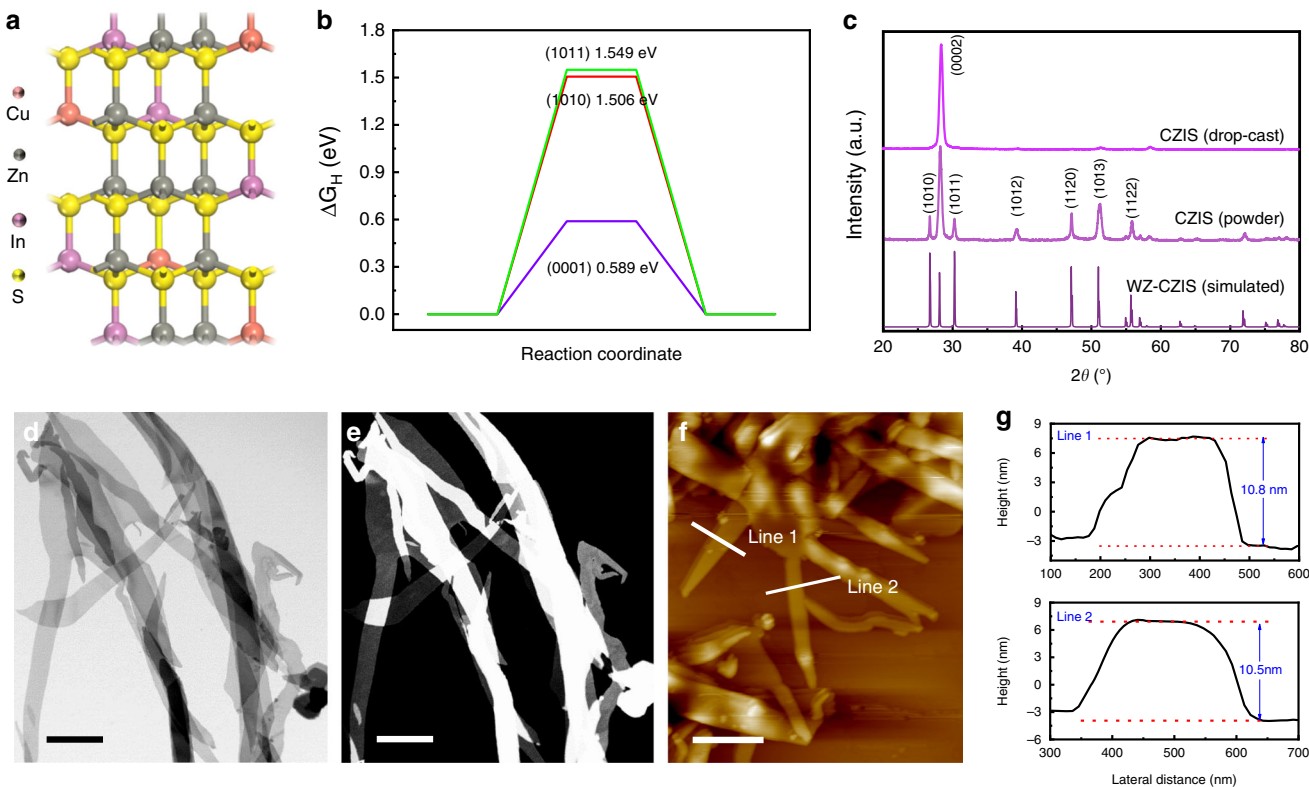

**Fig. 1 DFT calculation and two-dimensional morphology of CZIS nanobelts. a** The simulated crystal structure model. **b** Reaction Gibbs energy diagram for $H_2$ evolution of (0001), (1010), and (1011) crystal facets, respectively. **c** XRD patterns of the obtained CZIS nanobelts. The simulated wurtzite XRD pattern of CZIS is shown below for reference. **d**, **e** TEM and HAADF-STEM images of CZIS nanobelts. **f**, **g** AFM image and corresponding height images of the obtained wurtzite CZIS nanobelts. Scale bars are 500 nm for **d** and **e**, 1 μm for **f**, respectively.

2D structure with smooth surface and have a typical thickness of ca. 10 nm.

Figure 2 shows the further characterization of the obtained CZIS NBs. Figure 2a shows a typical CZIS nanobelt. The energy dispersive X-ray spectroscopy (EDS) spectrum shows the coexistence of Cu, Zn, In, and S elements (Supplementary Fig. 10), whereas the EDS-mapping and EDS-line analyses (Fig. 2c, d) demonstrate that these elements homogeneously distribute throughout the NBs. In addition, the high-resolution transmission electron microscope (HRTEM) images (Fig. 2e, f) in the top view of the randomly selected CZIS nanobelt show a perfect hexagonal lattice structure with a distinct lattice fringe of 0.34 nm, corresponding to the (1010) crystallographic plane of wurtzite CZIS. The selected area electron diffraction patterns (SAED) in Fig. 2g–i present a clear 6-fold symmetry bright plots, which correspond to hexagonal wurtzite structure oriented along [0001] plane, certifying the single crystalline nature and only the (0001) facet exposing of the obtained CZIS NBs. Furthermore, this colloidal method can be used to produce a series of wurtzite CZIS NBs with different Zn contents which expose (0001) facet (Supplementary Figs. 10–12 and Supplementary Table 2).

**General synthesis of single crystalline wurtzite CZGS NBs**. The suitability of the ligand assisted colloidal approach as a common method for synthesizing other 2D copper-based quaternary sulfides, i.e., CZGS, which is another important photocatalyst[15,50,51], has also been proved. The XRD pattern in Supplementary Fig. 13 demonstrated that the synthesized CZGS NBs have a wurtzite phase with an exposed (0001) facet. The HADDF-STEM and TEM images in Fig. 3a, b display the 2D nanobelt structure of the synthesized CZGS NBs. The AFM image shows the ultrathin

structure of CZGS NBs with a smooth surface and a thickness of ca. 9 nm (Supplementary Fig. 14). Figure 3c, d show the HRTEM images where the interplanar crystal spacing of 0.31 nm is index to (1010) plane of wurtzite CZGS. The SAED patterns (Fig. 3e and Supplementary Fig. 15) of one part in a CZGS nanobelt prove their single-crystal nature orientated along [0001] direction. Moreover, the EDS spectra (Supplementary Fig. 16), EDS-mapping and EDS-line scan analysis (Fig. 3f, g) of one part of the randomly selected CZGS nanobelt confirm the existence and homogeneous distribution of Cu, Zn, Ga, and S elements. Importantly, wurtzite CZGS NBs with the exposed (0001) facet that have different Zn contents can be obtained through this colloidal method (Supplementary Figs. 16–18 and Supplementary Table 5).

**Optical and photocatalytic properties**. Efficient absorption of sunlight is fundamental for solar-driven photocatalytic reactions. To investigate the absorbing capability of sunlight, thus, the absorption spectra of the synthesized NBs with different Zn content are collected by diffuse reflectance ultraviolet–visible–near-infrared (UV–vis–NIR) spectroscopy. As shown in Fig. 4a, b, CZIS and CZGS nanobelts exhibit a significant absorption in visible region with increased absorption region with decreased Zn content. The band gaps of CZIS and CZGS NBs have increased from 1.55 to 2.37 eV and 2.19 to 2.54 eV with Zn content (Supplementary Fig. 19 and Supplementary Tables 2–3), respectively. The obtained NBs exhibiting enough capability for harvesting solar light should be active photocatalysts under visible light. XPS valence band (VB) spectra were measured to confirm the relative locations of VB maximum of the obtained nanobelts, indicating that the CZIS and CZGS NBs are promising visible-light-driven photocatalysts (Supplementary Fig. 20).

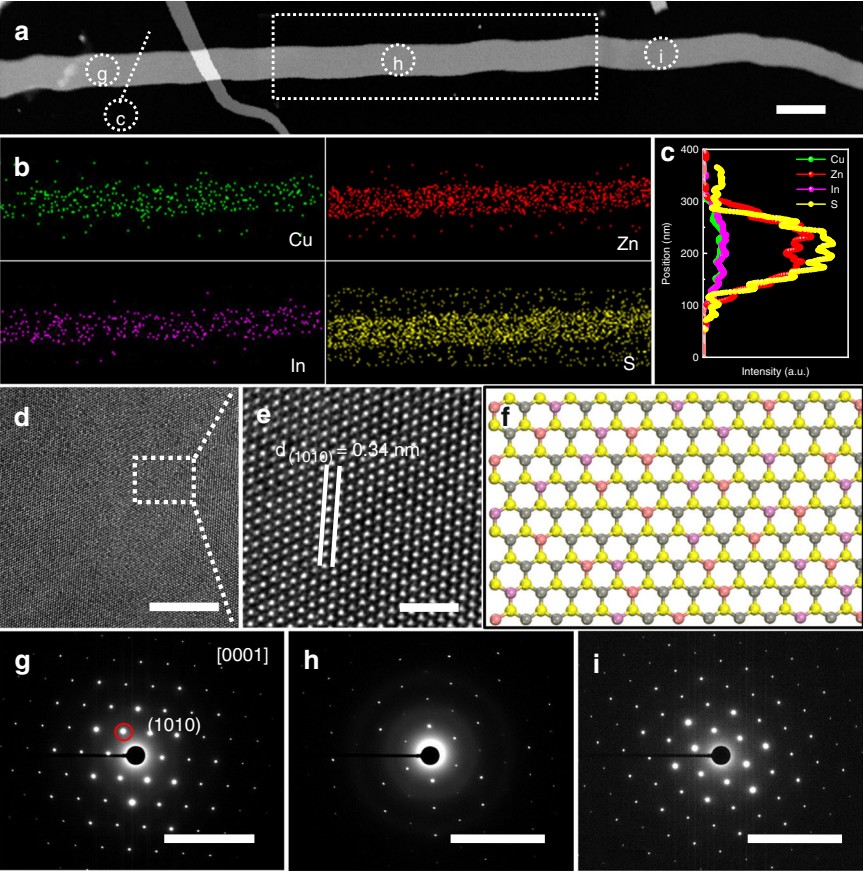

**Fig. 2 Characterization of the synthesized wurtzite CZIS nanobelts. a** HAADF-STEM image of a typical CZIS nanobelt. **b** EDS mapping of the selected part in **a**. **c** Smoothing simulation EDS-line scan analysis of the selected area in **a**. **d, e** HRTEM images. **f** Crystal model of (0001) facet of wurtzite CZIS nanobelts. **g–i** SAED pattern of the selected parts in **a**. Scale bars are 200 nm for **a**, 10 nm for **d**, 2 nm for **e**, 10 1/nm for **g–i**, respectively.

The photocatalytic hydrogen production performance of the synthesized NBs with different Zn contents was evaluated in a Pyrex reaction cell under Xe lamp irradiation ($\lambda \geq 420$ nm). No hydrogen was detected without irradiation. As shown in Fig. 4c (Supplementary Fig. 21a), the CZIS NBs display a Zn content dependent photocatalytic hydrogen evolution performance and reach the highest hydrogen production rate of 3.35 mmol h$^{-1}$ g$^{-1}$, which obviously exceeds the previously reported CZIS photocatalysts with similar compositions (Supplement Table 1). Furthermore, we measured the photocatalytic performance of the CZIS nanobelts with increasing amounts from 10 mg to 50 mg. The results (Supplementary Fig. 22) show that all the samples have the similar photocatalytic performance. In addition, wurtzite CZIS nanocrystals with different ratio of the (0001) facet have been synthesized to study the photocatalytic performance (Supplementary Figs. 23–25). The results revealed that CZIS nanocrystals with the most ratio of the (0001) facet had the best photocatalytic performance (Fig. 4d), proving that designing 2D structure CZIS exposed the (0001) facet can effectively enhance the photocatalytic property. Whereas, the CZGS nanobelt catalysts exhibit the highest hydrogen production rate of 3.75 mmol h$^{-1}$ g$^{-1}$ (Fig. 4e). We then tested the apparent quantum efficiencies (AQE) on these photocatalysts at diverse light wavelengths in the same reaction solution. The trend in AQEs are similar to that of the absorption spectra (Supplementary Fig. 26), indicating the bandgap-transition-dependent hydrogen production behavior.

Furthermore, the photocatalytic stabilities of CZIS and CZGS NBs were further evaluated (Fig. 4f) by long-term reaction. After each run, the irradiation was stopped and the reactor was evacuated before the next run, and no addition of fresh solution

and catalyst was conducted. The hydrogen evolution rates of CZIS and CZGS have almost no decreasing after six cycles. The morphology and structure of the nanobelt photocatalysts remained unchanged (Supplementary Figs. 27–29) and no obvious deactivation was observed, indicating good irradiation stability of these nanobelt photocatalysts. In addition, the CZIS NBs keep high stability and catalytic reactivity after storing two months (Supplementary Figs. 30–32). These results prove the excellent photocatalytic hydrogen evolution activity for the CZIS and CZGS NBs.

## Discussion

The synthesized CZIS and CZGS NBs exhibited excellent photocatalytic hydrogen production performance (Fig. 4g). The photoelectrochemical (PEC) experiments were further carried out to obtain more insight into optoelectronic properties of the synthesized nanobelts. The PEC results demonstrated distinct transient cathodic photocurrent response under visible-light irradiation (Supplementary Fig. 33). The current densities are higher than the recently reported copper-based quaternary sulfides due to the 2D structure and exposed facet in promoting the transportation and separation of photogenerated electrons and holes[14,51–53].

As show in Fig. 4c, d, the photocatalytic activities of the synthesized NBs relied on the composition. The dependence of the photocatalytic activities upon Zn content is mainly because of the change of the bandgap structure[17,54]. The photocatalytic activity of synthesized NBs increased with Zn content, indicating that the potential of conduction band is high enough to reduce water to

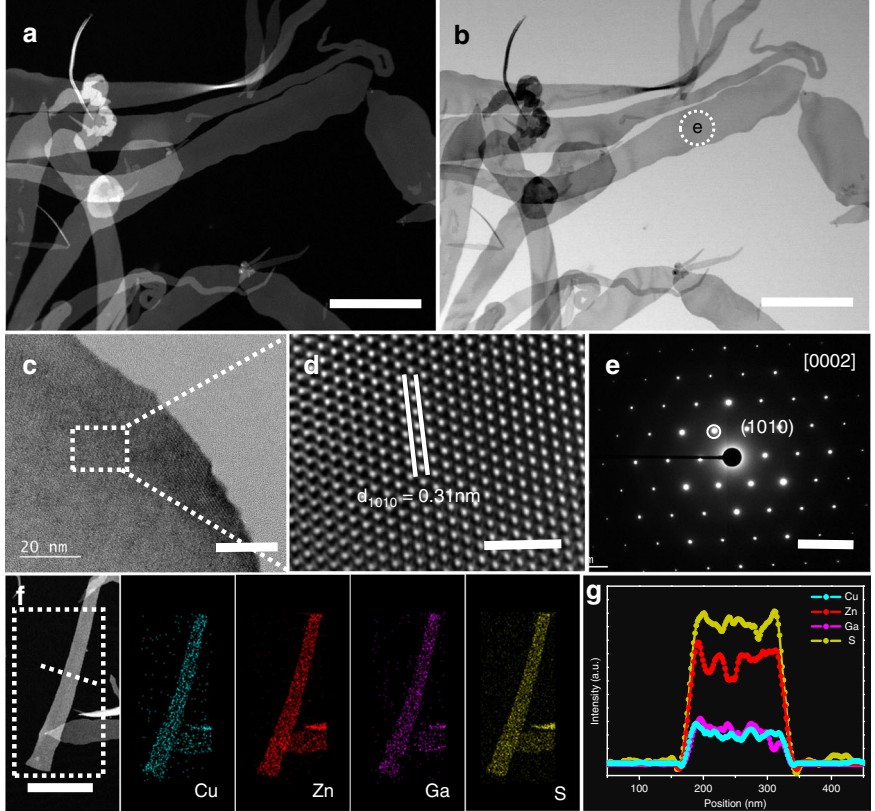

**Fig. 3 Characterization of the synthesized wurtzite CZGS nanobelts. a**, **b** HAADF-STEM and TEM micrographs. **c** HRTEM image. **d** Enlarged HRTEM image of the selected area in **c**. **e** SAED pattern of one part in **b**. **f** EDS mapping of one part of the randomly selected CZGS nanobelt. **g** Smoothing simulation EDS-line scan analysis of the selected area in **f**. Scale bars are 500 nm for **a**, **b**, 20 nm for **c**, 2 nm for **d**, 5 1/nm for **e**, 500 nm for **f**, respectively.

hydrogen (Supplementary Fig. 20). However, the activity for hydrogen evolution decreased when the Zn content exceeded four times. One reason for the activity decrease is that the absorption bands consisting of Cu and In were discrete as the content of Zn is much high, leading to the decrease in the mobility of photo-induced electrons and holes, which is an important factor for the photocatalytic performance[17]. Another possible factor for the low activity is that the decrease in the number of available photons with the extension of a band gap[54].

In summary, we have designed a potential kind of photo-catalysts, i.e., single-crystalline wurtzite CZIS nanobelts, which are composed of non-toxic and earth abundant elements and have high absorption efficiency in visible region. Firstly, we used DFT calculations to assess the photocatalytic hydrogen reaction Gibbs energy of different crystal facets of wurtzite CZIS. And then, we explored a colloidal method to synthesize single crystalline CZIS and CZGS NBs with the exposed (0001) facet that has the lowest reaction Gibbs energy. The obtained nanobelts exhibited excellent photocatalytic hydrogen production performance with the highest evolution rates of 3.35 and 3.75 mmol h$^{-1}$ g$^{-1}$ for CZIS and CZGS, respectively. In addition, the nanobelt photocatalysts show high stability and catalytic reactivity preservation after storing over 2 months. We anticipate that this photocatalyst design method can be exploited to other semiconductor material systems, thereby enabling novel photocatalysts that use the low-cost elements to efficiently catalyze special reactions.

## Methods
**Synthesis of single crystalline wurtzite CZIS NBs**. Cu(dedtc)$_2$ (0.2 mmol), In (dedtc)$_3$ (0.2 mmol), and Zn(dedtc)$_2$ (0.8 mmol) were added to a three-neck flask with in a mixed solvent (8 mL of OLA, 8 mL of DDT, and of 4 mL of ODE) in air.

Firstly, the reaction solution was heated up to 100 °C and degassed at this temperature for 20 min. Then, the reaction temperature rose to 250 °C at a heating rate of 10 °C/min and remained at 250 °C for 1 h under pure N$_2$. Lastly, the reaction solution naturally cooled down. The synthesized CZIS NBs were obtained by centrifuging. The obtained NBs were washed with hexane and ethanol for twice and dispersed in hexane. CZIS nanobelts with different Zn contents were synthesized using diversity amounts Zn(dedtc)$_2$ under the same reaction conditions (Supplementary Table 2).

**Synthesis of single crystalline wurtzite CZGS NBs**. The synthesis procedure of CZGS nanobelts is in the same with that of CZIS nanobelts with In(dedtc)$_3$ being substituted by Ga(dedtc)$_3$ (Supplementary Table 3).

**Measurement and characterization**. The X-ray power diffraction (XRD) of obtained nanobelts were characterized by using a Philips X'Pert PRO SUPER X-ray diffractometer which was equipped with graphite monochromatized Cu Kα radiation (λ = 1.54056 Å). The operation voltage was kept at 40 KV, and the current was kept at 400 mA. Diamond 3.2 was used to simulate the wurtzite XRD patterns of CZIS and CZGS. Nanobelts dispersed in hexane were dropped on Mo grid for transmission electron microscope (TEM), high-resolution transmission electron microscope (HRTEM) and high-angle annular dark field scanning transmission electron microscopy (HAADF-STEM) investigation observation, which were characterized using JEM-ARM200F with an acceleration voltage of 200 KV. Energy dispersive X-ray spectrometer (EDS) with mapping and line-scan modes was carried out on Inca Oxford equipped on JRM-ARM200F. DUV-3700 UV–vis–NIR spectrometer (Shimadzu) was used to investigated the optical properties of the synthesized NBs. XPS was collected on an ESCALab MKII X-ray photoelectron spectrometer using Mg Ka radiation exciting source. Tapping-mode atomic force microscopy (AFM) images were performed using a DI Innova Multimode SPM platform.

**Photoelectrochemical experiment**. The electrochemical measurements were carried out in a three-electrode cell with a counter electrode (platinum film) and a reference electrode (Ag/AgCl electrode). The active area of the working electrode was 2 cm². Al electrochemical experiments were performed in 0.1 M Eu(NO$_3$)$_3$ electrolyte. The photocurrent measurements were performed on electrochemical station (CH Instruments, model CHI 760E, Shanghai Chenhua Limited, China)

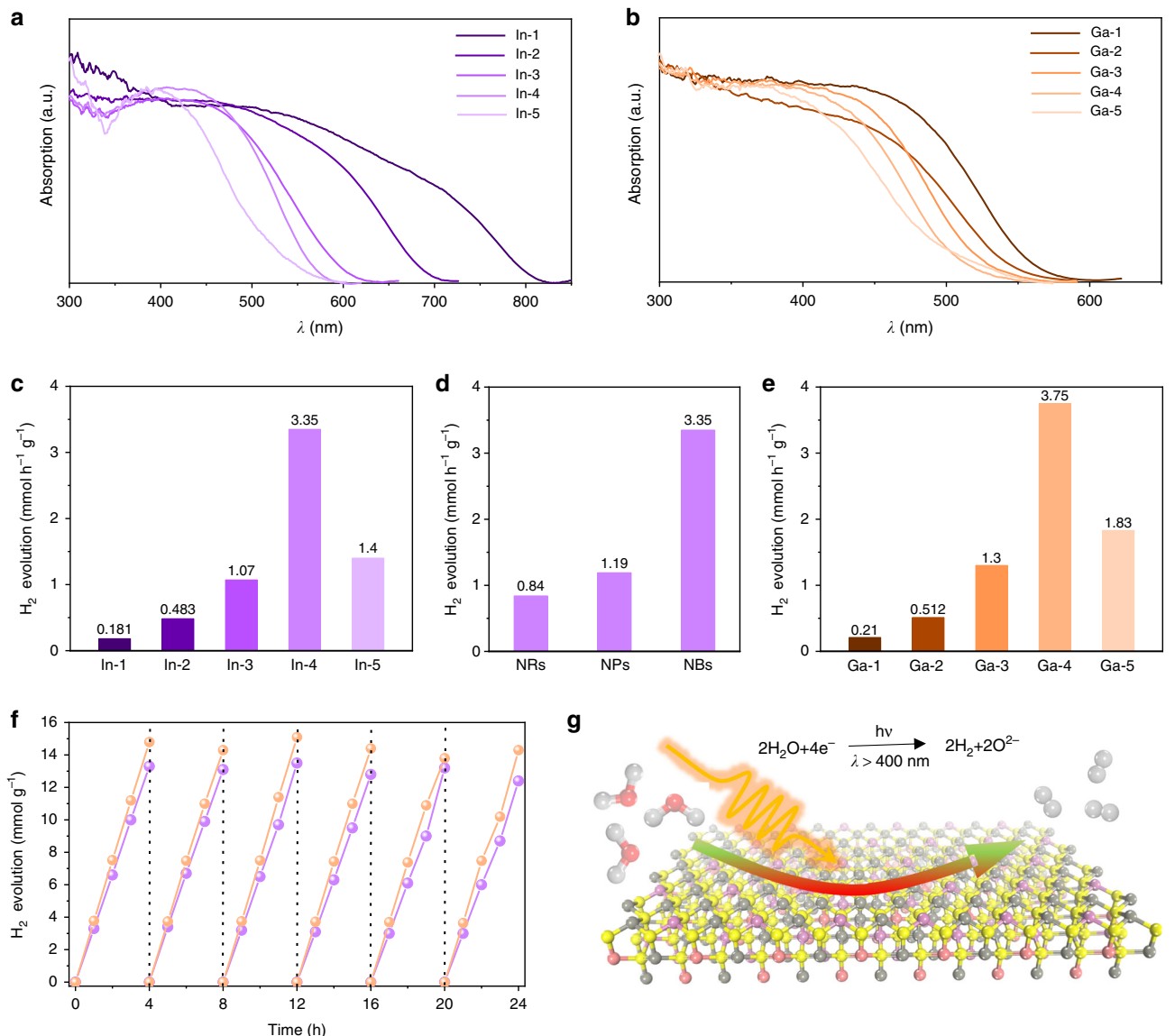

**Fig. 4 Optical and photocatalytic performances of the synthesized wurtzite CZIS and CZGS nanobelts. a, b** The UV-Vis-IR absorption spectra of CZIS and CZGS nanobelts, respectively. **c** Comparison of hydrogen production rates using CZIS nanobelt photocatalysts with different Zn contents under visible-light irradiation ($\lambda > 420$ nm). **d** The photocatalytic hydrogen evolution rates of CZIS nanoparticles, nanorods, and nanobelts with the same composition, respectively. **e** Comparison of hydrogen evolution rates of wurtzite CZGS nanobelt photocatalysts. **f** Recycle hydrogen generation property of CZIS and CZGS nanobelt photocatalysts. **g** Schematic diagram of the photocatalytic hydrogen evolution.

with simulated solar light irradiation by a 300W Xe lamp (PLS-SXE300C, Trusttech Co., Ltd. Beijing) equipped with an AM1.5 filter.

**Photocatalytic hydrogen evolution experiments**. The photocatalytic hydrogen evolution experiments were tested in a Pyrex reaction cell (Labsolar-IIIAG Photocatalytic system from PerfectLight Corporation). Typically, the obtained nanobelt catalyst (30 mg) was suspended in 100 mL of aqueous solution (0.25 M $Na_2SO_3$ and 0.35 M $Na_2S$), and subsequently sonicated for 30 min. The reaction solution was evacuated three times to remove air completely before irradiation under a 300W Xe lamp (Newport Corporation) which is equipped with a 420-nm cutoff filter. Meanwhile, the reaction solution was maintained at room temperature. The amount of hydrogen produced from the photocatalytic reaction was determined using a gas chromatograph (Agilent 7890A). The apparent quantum efficiency (AQE) was carried out under the same photocatalytic reactions by using 420 nm, 435 nm, 450 nm, 475 nm, 500 nm, 520 nm, 550 nm, and 600 nm band pass filters. The irradiation meter was used to calibrate the light intensity. The AQE was calculated by Eq. (1):

$$AQE = 2N_H/N_p \times 100\% \tag{1}$$

where $N_H$ represents the number of $H_2$ molecules and $N_P$ represents the number of incident photons.

**DFT calculations**. The crystallographic unit cell was a hexagonal corundum structure with S and metal atoms. We carried out DFT calculation using the Vienna ab initio simulation package (VASP)[55]. For bulk structure relaxation, the exchange-correlation energy was described using the generalized gradient approximation (GGA) with the formalism of Perdew–Burke–Ernzerhofer (PBE)[56]. A kinetic energy cutoff of 450 eV was applied to the plane-wave basis set, and a $3 \times 6 \times 5$ Gamma k-point sampling was adopted. The convergence criteria for total energy and force were $10^{-5}$ eV and $10^{-2}$ eV/Å, respectively. For band gap calculation, we used the range-separated hybrid HSE06 functional which is known for substantially improving prediction of band gaps of semiconductors relative to PBE functional[57].

The electronic structure calculations for (0001), (1010), and (1011) facets were investigated within the framework of spin-polarized DFT as implemented in the VASP code[58]. A five layers slab with a ($1 \times 2$) unit cell and a plane-wave basis cutoff of 450 eV are used. Vacuum areas of 15 Å are employed between neighboring slabs to avoid the interaction between layers. During the structural optimization, the lower two layers of the slab are fixed whereas the upper are

relaxed. The convergence criteria of the total energy and force on atoms are set to $10^{-5}$ eV and $10^{-2}$ eV/Å, respectively. The solvent effect of water has been treated implicitly using the VASPsol program[59–61].

**Calculation details for Gibbs free energy changes on hexagonal after adsorbing hydrogen atoms.**

$$\Delta G_H = \Delta E_H + \Delta E_{ZPE} - T\Delta S_H \qquad (2)$$

$$\Delta E_H = \Delta E_{ad} - (E_{surface} + 0.5 \times E_{H2}) \qquad (3)$$

$\Delta G_H$ represents for Gibbs free energy changes after adsorbing hydrogen atoms; $\Delta E_H$ represents for the energy changes of hydrogen atoms; $\Delta E_{ZPE}$ represents for the zero-point energy change; $T$ represents for temperature; $\Delta S_H$ represents for the entropy change of hydrogen atom; $\Delta E_{ad}$ represents for the surface energy with adsorbing hydrogen atom; $E_{surface}$ represents for the surface energy before adsorbing hydrogen atoms; and $E_{H2}$ represents for the energy of hydrogen.

## Data availability

The data that support the findings of this study are available on request from the corresponding author (S.-H.Y.).

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

## Acknowledgements

This work was supported by National Natural Science Foundation of China (Grants 51732011, 21431006, and 21761132008), the Foundation for Innovative Research Groups of the National Natural Science Foundation of China (Grant 21521001), Key Research Program of Frontier Sciences, CAS (Grant QYZDJ-SSW-SLH036), and the Users with Excellence and Scientific Research Grant of Hefei Science Center of CAS (2015HSC-UE007). L.W. acknowledges the funding support from the China Postdoctoral Science Foundation (2017M622016 and 2017LH006). G.Z. acknowledges the support from the Ministry of Science and Technology of the People's Republic of China (No. 2018YFA0208702). The Supercomputing Center of University of Science and Technology of China is acknowledged for numerical calculations.

## Author contributions

L.W., T.T.Z., F.J.F., and S.H.Y. conceived the idea. S.H.Y. supervised the project. L.W. carried out the experiments, and wrote the paper. T.T.Z. revised the paper. Y.L. and G.Q.L. helped with the photocatalysis experiments. Q.W. and G.Z.Z. carried out the DFT calculation and analyzed the computational result. L.S. helped to characterize and analyze the crystal structure. All authors discussed the results and assisted during manuscript preparation.

## Competing interests

The authors declare no competing interests.
