## [Peer Review File · Nature Communications]

Reviewers' Comments:

Reviewer #1:

Remarks to the Author:

Review on Manuscript ID : NCOMMS-20-05899-T

Title : "Single crystalline nonlayered quaternary sulfide nanobelts for efficient solar-to-hydrogen conversion"

Authors: Liang Wu et al.

This work presents a combined theoretical and experimental protocol for efficient screening of novel photocatalysts for hydrogen production. DFT calculations are combined to a colloidal synthesis to investigate wurtzite Cu-Zn-In-S and Cu-Zn-Ga-S nanobelts exposing a single (0001) facet as potential candidates for splitting water into hydrogen.

Since I am a theoretician, only the DFT part of this work will be evaluated in this review.

In my opinion, the calculations performed in the computational part of the proposed combined experimental/theoretical approach are only routine calculations, which are in addition not clearly presented. Therefore, from a computational viewpoint, the proposed computational/experimental approach does not deserve publication in Nature Communications.

Some major concerns regarding the DFT part:

- the PBE functional has been chosen with a spin-polarized formalism. The authors should much better justify this choice. In particular, key aspects such as electron delocalisation error or self-interaction should all be briefly discussed to briefly justify this choice.

- the adsorption of an H atom has been performed on three distinct surface orientations: adsorption energies and activation energies are reported. What is the thickness of each of the slabs considered ? How are transition states obtained ?

- the synthesis has been performed in aqueous solution, while all DFT calculations have been performed in gas phase. I am afraid that this might affect dramatically the reported activation energies. Since the DFT part is limited to the adsorption of H on 3 surfaces, including solvation effects in the calculation (explicitly and/or implicitly) could have been easily done and would have greatly improve the quality of the reported calculations.

Minor points:

- abstract: line 6: « density function » should be « density functional »

- p5, l13: « DTF » should be « DFT »

Reviewer #2:

Remarks to the Author:

Authors report on the synthesis of CZIS and CZGS nanobelts and their performance as photocatalyst for hydrogen evolution in the presence of a sacrificial hole scavenger. The manuscript is well organized and written and shows some relatively interesting results. In particular, the synthesized catalysts show very high hydrogen evolution rates. While there is a notable number of previous publications detailing the synthesis of CZIS and CZGS nanostructures and demonstrating their use for hydrogen evolution, the present work provides a protocol, based on well-known strategies and ligands, to produce planar nanostructures with exposed [0001] facets. Authors associate the excellent hydrogen evolution performance with this predominant [0001] facet and use DFT calculations to support this association. The significance of the manuscript is fair but it should be improved before publication in Nature Communications. In particular, the hydrogen evolution rate is strongly dependent on the reactor parameters, volume, amount of catalyst, etc. Authors should provide more details on the experimental set-up, such as irradiation on the sample. Also measurements with different catalyst concentrations should be performed and reported. Additionally, authors should attempt to calculate

the photonic/energy efficiency of the process of conversion of solar light to hydrogen. As a minor point, the word "nonlayered" may be removed from the title, as there is no comment or reference on the manuscript that supports the importance of this concept on the reported performances. Also, can authors calculate the crystal domain size from the XRD patterns in figure 1c? there seems to be not much difference between the distinct crystallographic plans, which is odd taking into account the large size differences of the crystals in different directions.

Reviewer #3:

Remarks to the Author:

For obtaining an efficient photocatalyst, Yu's group first used DFT calculation to reveal the lowest activation energy of (0001) facets. Referring to this theory, they developed a colloidal method to synthesize nonlayered single crystalline CZIS and CZGS nanobelts with the exposed (0001) facets, which expectedly exhibited high hydrogen evolution rates of 3.35 and 3.75 mmol h⁻¹ g⁻¹ much better than most of other CZIS and CZGS photocatalysts. Moreover, the excellent photocatalytic performance can be maintained for at least two months and reuse for six cycles. This work is very important and authors provided sufficient experiments to support the interesting result, so I strongly recommend it to be published after a minor revision.

Some small issues are as follows:

- 1) Is it necessary to emphasize "nonlayered" structure of nanobelts? I do not recognize its importance in this work.
- 2) In Supplementary Fig. 1, H atoms should be indicated more clearly on the figures. The panel "f" is lost in the caption?
- 3) Supplementary Figs. 1-3 should be described more for easily understanding the statement "...reveal a best electron environment of the (0001) facet ...".
- 4) The supplementary display items should appear in manuscript as consistent with their sequences in Supplementary Information.
- 5) It will be better to draw the range of visible light in Fig. 4a,b. The spectra were measured by using same concentration/amount of samples?
- 6) Language should further be checked for grammar, wording and typos. For example, line 7 and 15 on page 3; line 15 on page 5; line 7, 10 and 14 on page 6; line 7 on page 8; line 13 on page 9; line 17 on page 11, and so on.

Answers to the reviewers' questions

We have carefully considered all the concerns of the three independent reviewers, and have made suitable revision accordingly. For clearness reason, the answers were marked with RED color and started with “**” and the revision parts in the revised manuscript were marked in RED colour.

Reviewer #1 (Remarks to the Author):

This work presents a combined theoretical and experimental protocol for efficient screening of novel photocatalysts for hydrogen production. DFT calculations are combined to a colloidal synthesis to investigate wurtzite Cu-Zn-In-S and Cu-Zn-Ga-S nanobelts exposing a single (0001) facet as potential candidates for splitting water into hydrogen.

Since I am a theoretician, only the DFT part of this work will be evaluated in this review.

In my opinion, the calculations performed in the computational part of the proposed combined experimental/theoretical approach are only routine calculations, which are in addition not clearly presented. Therefore, from a computational viewpoint, the proposed computational/experimental approach does not deserve publication in Nature Communications.

** We thank the referee for the comment that prompts us to improve the quality of manuscript. We have conducted supplementary calculations as suggested and carefully revised the manuscript, and sincerely hope that our revisions have addressed the referee's concerns.

For understanding the interesting parts and significance of the present work, we would like to summarize our main findings here: In this work, we achieve the state-of-the-art visible-light-driven hydrogen generation performance for cocatalyst-free CZIS and CZGS nanobelt photocatalysts. The as-prepared nanobelt photocatalysts show an excellent composition-dependent photocatalytic performance, to reach the highest hydrogen evolution rate of 3.35 and 3.75 mmol h⁻¹ g⁻¹ for CZIS and CZGS nanobelts under visible-light irradiation ($\lambda > 420$ nm), respectively, which is higher than the reported CZIS and CZGS photocatalysts. Importantly, the NBs show high stability and catalytic reactivity retention after storing over a few months that further attests to the integrity of the constructed 2D materials for prospective advanced applications.

Some major concerns regarding the DFT part:

1. The PBE functional has been chosen with a spin-polarized formalism. The authors should much better justify this choice. In particular, key aspects such as electron delocalisation error or self-interaction should all be briefly discussed to briefly justify this choice.

** The main reason for using spin-polarized formalism is that the CZIS model has a non-zero magnetic moment originated from unpaired electrons (*NIC Series*, 2006, 31, 419-445.). However, electron delocalization error or self-interaction error is not suitable for pure GGA functional. To solve this issue, we have now employed range-separated hybrid functional HSE06, which is known for substantially improving the prediction of band gaps for semiconductors (*Phys. Rev. B*. 2009, 79, 115126).

Table R1 (new Supplementary Tables 4). Lattice constants of CZIS.

	a_0 (Å)	b_0 (Å)	c_0 (Å)	$c_0/2a_0$	$\alpha^*\beta^*\gamma$
Experiment	3.878	3.878	6.394	0.824	90*90*120
Simulation	3.855	3.855	6.328	0.820	90*90*120

As seen from Table R1, the calculated lattice constants are in good agreement with these measured in experiment. Then the simulated XRD pattern based on the PBE calculated structure are also in line with experiment (Fig. 1c). Furthermore, the band gap predicted by HSE06 (2.0 eV) meets the same with the experimentally measured value (2.1 eV) (Fig. R1). These results validate the feasibility of the DFT method implemented in this work.

Figure R1 (new Supplementary Fig.2). Total density of states of CZIS calculated using HSE06.

We have added above discussions to the revised manuscript.

2. The adsorption of an H atom has been performed on three distinct surface orientations: adsorption energies and activation energies are reported. What is the thickness of each of the slabs considered? How are transition states obtained?

The thickness of each of slabs considered are as follows: 14.88 Å for (1011) facet, 14.93 Å for (1010) facet and 16.79 Å for (0001) facet. To facilitate the comparison, we used 15 Å as the unified value for all of them. We fix the bottom three layers to mimic the bulk property while doing optimization for the rest of layers on the top that models surface property, as shown in Figure R2.

Figure R2 (new Supplementary Fig. 1). Crystal facets of CZIS which is used for simulations.

As for the activation energies, we need to clarify that we did not explicitly compute the reaction barrier using transition state method. Instead, we adopted an alternative thermodynamics approach proposed by Nørskov and co-workers (*J Am Chem Soc.* **2005**, 127 (15), 5308-5309.). We compared different facets as catalysts by calculating the free energy of atomic hydrogen bonding to the catalyst. The Bell-Evans-Polanyi principle is valid for a chemical reaction that proceeds along the reaction coordinate over the transition state. We therefore conclude that ΔG_H is a good descriptor of materials that can catalyze hydrogen evolution.

3. The synthesis has been performed in aqueous solution, while all DFT calculations have been performed in gas phase. I am afraid that this might affect dramatically the reported activation energies. Since the DFT part is limited to the adsorption of H on 3 surfaces, including solvation effects in the calculation (explicitly and/or implicitly) could have been easily done and would have greatly improve the quality of the reported calculations.

** Thank the review for the suggestion that helps us to improve the quality of work. We have recalculated the energies with implicit solvent effect taken into account and updated the relevant data in the revised manuscript.

As shown in Figure R3, we obtained update diagram of the Gibbs-free energy changes for HER, as well as the details were demonstrated in Table R2. The calculation results evident that the (0001) facet has the lowest energy barrier for photocatalytic hydrogen production.

Figure R3 (new Fig. 1b). Schematic diagram of the Gibbs-free energy changes for HER.

Table R2 (new Supplementary Table 3). The energy of atomic hydrogen bonding to the catalyst.

Facets	ΔE_H (eV)	$T\Delta S_H$ (eV)	ΔE_{ZEP} (eV)	ΔG_H (eV)
(0001)	0.163	-0.2	0.226	0.589
(1010)	1.091	-0.2	0.215	1.506
(1011)	1.126	-0.2	0.223	1.549

$$\Delta G_H = \Delta E_H + \Delta E_{ZPE} - T\Delta S_H$$

We have added above discussion to the revised manuscript.

Minor points:

- abstract: line 6: «density function » should be «density functional »
- p5, 113: «DTF » should be «DFT »

We thank the referee for the kind reminding. We have corrected these mistakes in the revised manuscript.

Reviewer #2 (Remarks to the Author):

Authors report on the synthesis of CZIS and CZGS nanobelts and their performance as photocatalyst for hydrogen evolution in the presence of a sacrificial hole scavenger. The manuscript is well organized and written and shows some relatively interesting results. In particular, the synthesized catalysts show very high hydrogen evolution rates. While there is a notable number of previous publications detailing the synthesis of CZIS and CZGS nanostructures and demonstrating their use for hydrogen evolution, the present work provides a protocol, based on well-known strategies and ligands, to produce planar nanostructures with exposed [0001] facets. Authors associate the excellent hydrogen evolution performance with this predominant [0001] facet and use DFT calculations to support this association. The significance of the manuscript is fair but it should be improved before publication in Nature Communications.

In particular, the hydrogen evolution rate is strongly dependent on the reactor parameters, volume, amount of catalyst, etc. Authors should provide more details on the experimental set-up, such as irradiation on the sample. Also, measurements with different catalyst concentrations should be performed and reported. Additionally, authors should attempt to calculate the photonic/energy efficiency of the process of conversion of solar light to hydrogen. As a minor point, the word “nonlayered” may be removed from the title, as there is no comment or reference on the manuscript that supports the importance of this concept on the reported performances. Also, can authors calculate the crystal domain size from the XRD patterns in figure 1c? there seems to be not much difference between the distinct crystallographic plans, which is odd taking into account the large size differences of the crystals in different directions.

** We thank the referee for these helpful comments. The detail photocatalytic H₂ evolution experiments have been described in the Method of the revised manuscript. We now provide the supplementary experimental conditions. The photocatalytic experiments were performed in Labsolar-IIIAG Photocatalytic system (PerfectLight Corporation) with a 250 mL reactor. The used amount of the catalyst is 30 mg. A 300 W Xe lamp (Newport Corporation) equipped with UV cut-off filter (>420 nm) was used to irradiate the sample.

Photocatalytic H₂ evolution experiments with different catalyst concentrations have been measured under the same conditions. As shown in Figure R4, all catalysts with increasing amounts from 10 mg to 50 mg exhibit the similar photocatalytic performance.

Figure R4 (new Supplementary Fig. 21). (a) Comparison of hydrogen evolution rates under visible-light irradiation using different amounts of CZIS nanobelt photocatalysts. (b) Comparison of hydrogen evolution activities of CZIS nanobelt photocatalysts with different amounts.

The apparent quantum efficiency (AQE) was measured under the identical photocatalytic reactions by using 420 nm, 435 nm, 450 nm, 475 nm, 500 nm, 520 nm, 550 nm, and 600 nm band pass filters, respectively. The trend in apparent quantum efficiency closely followed that of the absorbance measured by ultraviolet–visible spectroscopy (Figure R5), revealing bandgap-transition-dependent hydrogen evolution behavior.

Figure R5 (new Supplementary Fig. 25). Photocatalytic efficiency of the synthesized NBs. Apparent quantum efficiency (AQE) in photocatalytic H₂ evolution of (a) CZIS NBs and (b) CZGS NBs.

Thanks for the referee's suggestion, we have now removed the word "nonlayered" from the title.

The crystal domain size is calculated from XRD patterns using Scherrer Formula:

$$D=K\gamma/B\cos\theta$$

K represents Scherrer constant; γ represents the X-ray wavelength; B is the half-maximum line breadth; θ is Bragg diffraction Angle; D represents the average thickness perpendicular to the facet.

The crystal size of (1010), (0001) and (1011) facets are 49.9 nm, 13nm, and 21.5 nm, respectively.

We have now added above discussions to the revised manuscript.

Reviewer #3 (Remarks to the Author):

For obtaining an efficient photocatalyst, Yu's group first used DFT calculation to reveal the lowest activation energy of (0001) facets. Referring to this theory, they developed a colloidal method to synthesize nonlayered single crystalline CZIS and CZGS nanobelts with the exposed (0001) facets, which expectedly exhibited high hydrogen evolution rates of 3.35 and 3.75 mmol h⁻¹ g⁻¹ much better than most of other CZIS and CZGS photocatalysts. Moreover, the excellent photocatalytic performance can be maintained for at least two months and reuse for six cycles. This work is very important and authors provided sufficient experiments to support the interesting result, so I strongly recommend it to be published after a minor revision.

Some small issues are as follows:

1) Is it necessary to emphasize “nonlayered” structure of nanobelts? I do not recognize its importance in this work.

** Thank you for your helpful suggestion. We have now removed the word “nonlayered” from the title.

2) In Supplementary Fig. 1, H atoms should be indicated more clearly on the figures. The panel “F” is lost in the caption?

** New figure is shown as follow.

Figure R6 (new Supplementary Fig. 3). The adsorption sites of H atom. a (1011) facet. b, (1010) facet. c, (0001) facet.

3) Supplementary Figs. 1-3 should be described more for easily understanding the statement "...reveal a best electron environment of the (0001) facet ...".

**** We have added the following statement to the main text:**

The H absorption energy of different sites in the selected facets (Supplementary Fig. 3 and Supplementary Table 3) reveal that the (0001) facet provide a favorable environment for hydrogen production.

4) The supplementary display items should appear in manuscript as consistent with their sequences in Supplementary Information.

**** We have changed the display items in the revised supplementary information.**

5) It will be better to draw the range of visible light in Fig. 4a, b. The spectra were measured by using same concentration/amount of samples?

**** The range of visible light is 400-800 nm. The spectra were measured using CZIS/CZGS films fabricated by dip-coating 0.5 mL of CZIS/CZGS hexane solution (20 mg/mL) on a 20*20*1 mm quartz glass.**

6) Language should further be checked for grammar, wording and typos. For example, line 7 and 15 on page 3; line 15 on page 5; line 7, 10 and 14 on page 6; line 7 on page 8; line 13 on page 9; line 17 on page 11, and so on.

**** We have corrected these language issues and carefully checked the grammar, wording and typos both in the manuscript and supplementary information.**

Reviewers' Comments:

Reviewer #1:

Remarks to the Author:

Review on Manuscript ID : NCOMMS-20-05899A

Title : "Single crystalline quaternary sulfide nanobelts for efficient solar-to-hydrogen conversion"

Authors: Liang Wu et al.

The authors have addressed most of my comments correctly. In my opinion, the computational details section still lacks proper references to a number of key aspects of the computational setup. In particular:

- a proper reference to the VASP code should be added
- a proper reference to the PBE functional should be added
- a proper reference to the HSE06 functional should be added
- the two original references to the VASPsol development work should be added:

Implicit solvation model for density-functional study of nanocrystal surfaces and reaction pathways.

K. Mathew, R. Sundararaman, K. Letchworth-Weaver, T. A. Arias, and R. G. Hennig.

J. Chem. Phys. 140, 084106 (2014).

Accuracy of Exchange-Correlation Functionals and Effect of Solvation on the Surface Energy of Copper.

M. Fishman, H. L. Zhuang, K. Mathew, W. Dirschka, and R. G. Hennig.

Phys. Rev. B 87, 245402 (2013)

Reviewer #2:

Remarks to the Author:

Authors properly took into account reviewer points. The manuscript shows is well written and shows interesting conclusion that are properly supported on experimental results and calculations. I recommend its publication in Nature Comm.

Manuscript ID: NCOMMS-20-05899A

“Single crystalline quaternary sulfide nanobelts for efficient solar-to-hydrogen conversion”

Reviewer #1 (Remarks to the Author):

The authors have addressed most of my comments correctly. In my opinion, the computational details section still lacks proper references to a number of key aspects of the computational setup.

In particular:

- a proper reference to the VASP code should be added
- a proper reference to the PBE functional should be added
- a proper reference to the HSE06 functional should be added
- the two original references to the VASPsol development work should be added:

Implicit solvation model for density-functional study of nanocrystal surfaces and reaction pathways. K. Mathew, R. Sundararaman, K. Letchworth-Weaver, T. A. Arias, and R. G. Hennig. J. Chem. Phys. 140, 084106 (2014).

Accuracy of Exchange-Correlation Functionals and Effect of Solvation on the Surface Energy of Copper. M. Fishman, H. L. Zhuang, K. Mathew, W. Dirschka, and R. G. Hennig. Phys. Rev. B 87, 245402 (2013).

****We thank the reviewer for the helpful suggestions. We have added proper references in the revised manuscript.**

Reviewer #2 (Remarks to the Author):

Authors properly took into account reviewer points. The manuscript is well written and shows interesting conclusion that are properly supported on experimental results and calculations. I recommend its publication in Nature Comm.

****We thank the reviewer for strong support on the publication of this work.**